# Regex Queries over Incomplete Knowledge Bases

**Vaibhav Adlakha**[*]                    VAIBHAVADLAKHA95@GMAIL.COM
*McGill University, MILA - Quebec AI Institute*

**Parth Shah**                    PARTHUSHAH8@GMAIL.COM
*Indian Institute of Technology Delhi*

**Srikanta Bedathur**                    SRIKANTA@CSE.IITD.AC.IN
*Indian Institute of Technology Delhi*

**Mausam**                    MAUSAM@CSE.IITD.AC.IN
*Indian Institute of Technology Delhi*

## Abstract

We propose the novel task of answering regular expression queries (containing disjunction ($\vee$) and Kleene plus ($+$) operators) over incomplete KBs. The answer set of these queries potentially has a large number of entities, hence previous works for single-hop queries in KBC that model a query as a point in high-dimensional space are not as effective. In response, we develop *RotatE-Box* – a novel combination of *RotatE* and *box* embeddings. It can model more relational inference patterns compared to existing embedding based models. Furthermore, we define baseline approaches for embedding based KBC models to handle regex operators. We demonstrate performance of *RotatE-Box* on two new regex-query datasets introduced in this paper, including one where the queries are harvested based on actual user query logs. We find that our final *RotatE-Box* model significantly outperforms models based on just *RotatE* and just box embeddings.

## 1. Introduction

Knowledge Base Completion (KBC) predicts unseen facts by reasoning over known facts in an incomplete KB. Embedding-based models are a popular approach for the task – they embed entities and relations in a vector space, and use a scoring function to evaluate validity of any potential fact. KBC models are typically evaluated using *single-hop* queries, example, "Who founded Microsoft?", which is equivalent to the query ($Microsoft \rightarrow founded\_by \rightarrow$ ?). A natural extension [Guu et al., 2015] poses *multi-hop* queries, such as ($Microsoft \rightarrow founded\_by \rightarrow \cdot \rightarrow lives\_in \rightarrow$?), which represents "Where do founders of Microsoft live?". Recent works have also studied subsets of *first-order logic queries* with conjunction ($\wedge$), disjunction($\vee$) and existential quantifiers ($\exists$) [Hamilton et al., 2018, Ren et al., 2020].

We analyzed real-world query logs[1] over the publicly available Wikidata KB[2] and extract its distribution of queries. We found (see Table 1) that while the single hop queries are indeed most frequent (87%), multi-hop queries comprise only 1% of all queries. A substantial fraction of queries (12%) are queries using *regular expression* operators (regex), which use disjunction ($\vee$) or Kleene plus ($+$) operators. In response, we pose the novel problem of answering regex queries using embedding-based KBC models.

---

[*]Work done as a research intern at IIT Delhi
[1]https://iccl.inf.tu-dresden.de/web/Wikidata_SPARQL_Logs/en
[2]https://www.wikidata.org/wiki/Wikidata:Main_Page

Regex queries have a potentially large set of correct answers, due to which typical embedding models that model both queries and entities as points are not very effective. We build on one such model *RotatE*, which models a relation as a rotation of entity vectors – it can model all relation patterns generally observed in a KB except hierarchy and has not been extended to queries beyond single-hop. There also exists Query2Box [Hamilton et al., 2018], a box embedding method which models a query as a high-dimensional axis-aligned box. This can better handle larger answer spaces, but it cannot model symmetric relations (e.g. *friend_of*).

| Query Type | %age in Query Log |
|---|---|
| Single Hop Queries | 86.98% |
| Multi-Hop Queries | 1.02% |
| Regex Queries | 11.98% |

Table 1: User queries in Wikidata logs

To answer the full repertoire of regex queries over incomplete KBs, we propose a new baseline, *RotatE-Box* – a RotatE model augmented with box embeddings. In our model, a regex query is an axis-aligned box. Application of a relation or regex expression rotates the center (vector) of the box, and enlarges/shrinks the box. Application of a regex operator is handled using two approaches: (1) learned operators over boxes, or (2) introducing separate embeddings for Kleene plus of a relation, and expanding a disjunction into multiple non-disjunctive queries. *RotatE-Box* can capture all typical relation patterns – symmetry, antisymmetry, inversion, composition, intersection, mutual exclusion and hierarchy, along with effectively handling a large number of correct answers.

| **FB15K** |
|---|
| Justin Timberlake, $(friend\|peers)^+$, ? |
| Avantgarde, $(parent\_genre)^+$, ? |
| Agnes Nixon, $place\_of\_birth/adjoins^+$, ? |

| **Wiki100** |
|---|
| Keanu Reeves, $place\_of\_birth\|residence$, ? |
| Donald Trump, $field\_of\_work/subclass\_of^+$, ? |
| Electronic Dance Music, $(instance\_of\|subclass\_of)^+$, ? |

Table 2: Example queries from FB15K and Wiki100

We also contribute two new datasets for our task. Our first dataset, called FB15K-Regex, is an extension of the popular FB15K dataset [Bordes et al., 2013], and is constructed using random walks on the FB15K graph. Our second dataset is built over a dense subset of Wikidata (which we name $Wiki100$) and is based on the real queries from existing query logs. Table 2 lists example queries in the datasets. We find that models based on *RotatE-Box* outperform all other baselines, including strong models based on just *RotatE* and just *Query2Box* on these two datasets. In summary, our work presents a novel regex query answering task, two new datasets and a strong baseline for our task. We will release all resources for further research[3].

## 2. Related Work

A KB $\mathcal{G}$ is a set of triples $(e_1, r, e_2)$, where $e_1, e_2 \in \mathcal{E}$ (the set of entities) and $r \in \mathcal{R}$ (the set of relations). For a KBC task, all triples are split into train/dev/test, resulting in graphs $\mathcal{G}_{train}$, $\mathcal{G}_{dev}$, and $\mathcal{G}_{test}$. The standard KBC task requires answering single-hop queries of the form $(e_1, r, ?)$ with the answer $e_2^*$ if $(e_1, r, e_2^*) \in \mathcal{G}_{test}$. While many kinds of solutions exist for this problem, we build on the vast literature in embedding-based approaches [Bordes

---

[3] https://github.com/dair-iitd/kbi-regex

et al., 2013, Socher et al., 2013, Yang et al., 2014, Trouillon et al., 2016, Jain et al., 2018, Sun et al., 2019, Jain et al., 2020]. These methods embed each entity and relation in a latent embedding space, and use various tensor factorization scoring functions to estimate the validity of a triple. Our work builds on RotatE, which defines relation as a rotation in a complex vector space. The distance of target entity from the query constitutes the scoring function. RotatE has shown high KBC performance and can model many relation patterns (see Section 2.1).

The first work on answering complex queries over incomplete KBs extends single-hop queries to path queries (or multi–hop queries) of the type $(e_1, r_1, r_2, \ldots, r_n, ?)$. Guu et al. [2015] construct datasets for the task using random walks on FB13 and WN11 KBs. Models for the task include compositional training of existing KBC scoring functions and models based on variants of RNNs [Yin et al., 2018, Das et al., 2017]. Recent work has started exploring subsets of first-order logic queries. Hamilton et al. [2018] explore conjunctive queries with conjunction ($\wedge$) and existential quantification ($\exists$). They propose DeepSets [Zaheer et al., 2017] to model conjunction as a learned geometric operation. MPQE [Daza and Cochez, 2020] and BIQE [Kotnis et al., 2021] extend this work by constructing a query representation using a graph neural network and transformer encoder respectively.

Probably closest to our work is that of answering Existential Positive First-order (EPFO) logical queries [Ren et al., 2020] – queries that include $\wedge$, $\vee$, and $\exists$. They propose Query2Box, which represents a query as a box (center, offset) and each entity as point in a high-dimensional space. The points closer to the center are scored higher for the query, especially if they are inside the box. The strength of Query2Box is that it can naturally handle large answer spaces, and also model many relation patterns (see Section 2.1). [Ren and Leskovec, 2020] extend EPFO to complete set of first-order logical operations, by handling negation ($\neg$) with their proposed model, *BetaE*. They propose high-dimensional Beta distributions as embeddings of queries and entities. The KL divergence between query and entity serves as a scoring function.

There are also other works that work with probabilistic databases for answering EPFO queries [Friedman and Van den Broeck, 2020], but, to the best of our knowledge, there has been no work on answering *regex* queries over incomplete knowledge bases. In particular, regular expressions that contain the Kleene plus operator, are not included in path queries or first-order logic queries.

## 2.1 Relation Patterns in a KB

Embedding models can be critiqued based on their representation power [Toutanova and Chen, 2015, Guu et al., 2015, Trouillon et al., 2016, Sun et al., 2019] in modeling commonly occuring relation patterns (relationships amongst relations). We summarize the common patterns and model capabilities in Table 3. Broadly, translation-based methods (that add and subtract embedding vectors) like TransE and Query2Box can model all patterns but cannot model symmetry ($(e_1, r_1, e_2) \Rightarrow (e_2, r_1, e_1)$), such as for the relation *friend_of*. Rotation-based models such as RotatE can model all inference patterns except for hierarchy: $(e_1, r_1, e_2) \Rightarrow (e_1, r_2, e_2)$. Other models have other weaknesses, like similarity-based methods (that use dot products between embeddings) such as DistMult and ComplEx

| Pattern | Formula | TransE | DistMult | ComplEx | RotatE | Query2Box | RotatE-Box |
|---------|---------|--------|----------|---------|--------|-----------|------------|
| Symmetry | $r(x,y) \Rightarrow r(y,x)$ | ✗ | ✓ | ✓ | ✓ | ✗ | ✓ |
| Antisymmetry | $r(x,y) \Rightarrow \neg r(y,x)$ | ✓ | ✗ | ✓ | ✓ | ✓ | ✓ |
| Inversion | $r_1(x,y) \Rightarrow r_2(y,x)$ | ✓ | ✗ | ✓ | ✓ | ✓ | ✓ |
| Composition | $r_1(x,y) \wedge r_2(y,z) \Rightarrow r_3(x,z)$ | ✓ | ✗ | ✗ | ✓ | ✓ | ✓ |
| Hierarchy | $r_1(x,y) \Rightarrow r_2(x,y)$ | ✗ | ✓ | ✓ | ✗ | ✓ | ✓ |
| Intersection | $r_1(x,y) \wedge r_2(x,y) \Rightarrow r_3(x,y)$ | ✓ | ✗ | ✗ | ✓ | ✓ | ✓ |
| Mutual Exclusion | $r_1(x,y) \wedge r_2(x,y) \Rightarrow \perp$ | ✓ | ✓ | ✓ | ✓ | ✓ | ✓ |

Table 3: The relation patterns modeling capabilities of several embedding-based KBC methods. RotatE-Box models more relation patterns than any other method.

are incapable of handling composition and intersection. Our proposed model combines the power of Query2Box and RotatE and can express all seven common relation patterns.

## 3. Task

We now formally define the task of answering regular expressions (regex) queries on a KB. Regular expressions are primarily used to express navigational paths on a KB in a more succinct way and to extend matching to arbitrary length paths. Any regex $c$ in a KB is an expression over $\mathcal{R} \cup \{+, /, \vee\}$, generated from the grammar: $c ::= r \mid c_1 \vee c_2 \mid c_1/c_2 \mid c^+$, where $r \in \mathcal{R}$ represents a relation in the KB and $c_1$ and $c_2$ represent regular expressions. $/$ denotes *followed by*, $\vee$ denotes *disjunction*, and $+$ is a *Kleene Plus* operator denoting one or more occurrences. We use $l(c)$ to denote the set of paths (sequence of relations) compatible with regex $c$. As an example, for $c = r_1/r_2^+$, $l(c)$ will have paths like $r_1/r_2$, $r_1/r_2/r_2$, $r_1/r_2/r_2/r_2$ and so on. Their path lengths will be 2, 3 and 4 respectively.

A regex query $q$ over a KB is defined as a pair of head entity $e_1$ and a regular expression $c$. For query $q = (e_1, c, ?)$, we define the answer set $[\![q]\!]$ as the union of entities reachable from $e_1$ in $\mathcal{G}$, by following the paths in $l(c)$: $[\![q]\!] = \bigcup_{p \in l(c)}[\![(e_1, p, ?)]\!]$. The goal of the task is to, given a query $q$, rank entities in $[\![q]\!]$ higher than other entities.

## 4. Model

We now describe *RotatE-Box* – a model that combines the strengths of RotatE and Query2Box. Following Query2Box, the relations and queries are modeled as boxes, and entities as points in a high-dimensional space. However, unlike Query2Box, all points (and boxes) are in a complex vector space $\mathbb{C}^k$, instead of a real space.

We first describe the model for single-hop queries. RotatE-Box embeds each entity $e$ as a point $\mathbf{e} \in \mathbb{C}^k$ and each relation $r$ as a box $\mathbf{r}$, represented as $(\text{Cen}(\mathbf{r}), \text{Off}(\mathbf{r})) \in \mathbb{C}^{2k}$. Here $\text{Cen}(\mathbf{r}) \in \mathbb{C}^k$ is the center of the box such that its modulus in each of the $k$ dimensions is one, i.e., $\forall j \in [1, k], |\text{Cen}(r_j)| = 1, \text{Cen}(r_j) \in \mathbb{C}$. By doing so, $\text{Cen}(r_j)$ can be re-written as $e^{i\theta_{r,j}}$ using Euler's formula (here, $e$ is Euler's number). This corresponds to rotation by angle $\theta_{r,j}$ in the $j^{th}$ dimension of complex vector space. In vectorized form, $\text{Cen}(\mathbf{r}) = e^{i\boldsymbol{\theta}_r}$. $\text{Off}(\mathbf{r})$ is a positive offset, i.e., $\text{Re}(\text{Off}(\mathbf{r})) \in \mathbb{R}_{\geq 0}^k$ and $\text{Im}(\text{Off}(\mathbf{r})) \in \mathbb{R}_{\geq 0}^k$. Any box that satisfies these constraints on its center and offset is termed as a *rotation box*. A point $\mathbf{v} \in \mathbb{C}^k$ is considered

*inside* a box if it satisfies:

$$\text{Re}(\mathbf{r}_{min}) \leq \text{Re}(\mathbf{v}) \leq \text{Re}(\mathbf{r}_{max}) \quad \text{and} \quad \text{Im}(\mathbf{r}_{min}) \leq \text{Im}(\mathbf{v}) \leq \text{Im}(\mathbf{r}_{max}) \tag{1}$$

Here, $\mathbf{r}_{max} = \text{Cen}(\mathbf{r}) + \text{Off}(\mathbf{r}) \in \mathbb{C}^k$, and $\mathbf{r}_{min} = \text{Cen}(\mathbf{r}) - \text{Off}(\mathbf{r}) \in \mathbb{C}^k$. The inequalities represent element-wise inequality of vectors. A single-hop query $q = (e_1, r, ?)$ has a rotation box representation $\mathbf{q}$ given by $(\mathbf{e}_1 \odot \text{Cen}(\mathbf{r}), \text{Off}(\mathbf{r}))$. Here $\odot$ is the Hadamard product (element-wise multiplication). Since $\text{Cen}(\mathbf{r}) = e^{i\boldsymbol{\theta}_r}$, $\mathbf{e}_1$ is rotated by an angle $\boldsymbol{\theta}_r \in \mathbb{R}^k$. Specifically $e_j$ is rotated by angle $\theta_{r,j}$ in the $j^{th}$ dimension. Given a query $q$ associated with a box $\mathbf{q}$, RotatE-Box ranks entities $e$ in the order of the distance of $\mathbf{e}$ from $\mathbf{q}$, defined as:

$$\text{dist}(\mathbf{e}; \mathbf{q}) = \text{dist}_{\text{out}}(\mathbf{e}; \mathbf{q}) + \alpha \cdot \text{dist}_{\text{in}}(\mathbf{e}; \mathbf{q}) \tag{2}$$

where $\alpha \in (0, 1)$ is a constant to downweigh the distance inside the query box, and

$$\begin{aligned}
\text{dist}_{\text{out}}(\mathbf{e}; \mathbf{q}) &= \|\text{Re}(\text{Max}(\mathbf{e} - \mathbf{q}_{max}, \mathbf{0}) + \text{Max}(\mathbf{q}_{min} - \mathbf{e}, \mathbf{0}))\|_1 \\
&+ \|\text{Im}(\text{Max}(\mathbf{e} - \mathbf{q}_{max}, \mathbf{0}) + \text{Max}(\mathbf{q}_{min} - \mathbf{e}, \mathbf{0}))\|_1 \\
\text{dist}_{\text{in}}(\mathbf{e}; \mathbf{q}) &= \|\text{Re}(\text{Cen}(\mathbf{q}) - \text{Min}(\mathbf{q}_{max}, \text{Max}(\mathbf{q}_{min}, \mathbf{e})))\|_1 \\
&+ \|\text{Im}(\text{Cen}(\mathbf{q}) - \text{Min}(\mathbf{q}_{max}, \text{Max}(\mathbf{q}_{min}, \mathbf{e})))\|_1
\end{aligned}$$

Here $\mathbf{q}_{max}$ and $\mathbf{q}_{min}$ are defined analogously to $\mathbf{r}_{max}$ and $\mathbf{r}_{min}$ above.

This completes the description of RotatE-Box for single-hop queries. As RotatE is a special case of RotatE-Box (by modeling $\text{Off}(\mathbf{r}) = \mathbf{0}, \forall r \in \mathcal{R}$), RotatE-Box can model all the relation patterns modeled by RotatE: symmetry, anti-symmetry, inversion, composition, intersection and mutual exclusion. Additionally, boxes enable modeling hierarchy.

**Theorem 1.** *RotatE-Box can model hierarchical relation pattern. (Proof in Appendix A.1)*

### 4.1 Relation Paths

We now define RotatE-Box operations for a path query, e.g., $q = (e_1, r_1/r_2/\ldots/r_n, ?)$. It first initializes the rotation box embedding for relation path as $\mathbf{r}_1$ (denoted by $\mathbf{p}_1$). Given $\mathbf{p}_i$, to continue the path with $r_{i+1}$, it rotates the center of $\mathbf{p}_i$, and adds the offsets:

$$\text{Cen}(\mathbf{p}_{i+1}) = \text{Cen}(\mathbf{p}_i) \odot \text{Cen}(\mathbf{r}_{i+1}) \qquad \text{Off}(\mathbf{p}_{i+1}) = \text{Off}(\mathbf{p}_i) + \text{Off}(\mathbf{r}_{i+1}) \tag{3}$$

The final rotation box embedding $\mathbf{p}$ of $r_1/r_2/\ldots/r_n$ is $\mathbf{p}_n$ as computed above. It computes query embedding $\mathbf{q}$ as described earlier – rotating $\mathbf{e}_1$ by $\text{Cen}(\mathbf{p})$ and keeping the offset as $\text{Off}(\mathbf{p})$. Ranking of answers for the query follows similar approach as specified in Eq 2.

### 4.2 Kleene Plus

To extend path embeddings to general regex expression $c$, we follow the same representation: for a regex query $(e_1, c, ?)$, $c$ is embedded as a rotation box, $\mathbf{c}$, in complex space $\mathbb{C}^{2k}$. We first describe our formulation for the Kleene Plus $(+)$ operator. We use two methods for this.

**Projection:** Kleene Plus $(+)$ operator may apply compositionally and recursively to any regular expression. An approach to materialize this is to define it as a geometric operator

[Hamilton et al., 2018], which takes input a rotation box representation $\mathbf{c}$ (corresponding to rotation by $\boldsymbol{\theta}_c$), applies a function $kp$ to output a new rotation box embedding $\mathbf{c}'$ that represents $c^+$. RotatE-Box uses the function $kp(\mathbf{c}) = \mathbf{c}' = (e^{i\boldsymbol{\theta}_{c'}}, \mathbf{K}_{\text{off}}\text{Off}(\mathbf{c}))$, where $\boldsymbol{\theta}_{c'} = \mathbf{K}_{\text{cen}}\boldsymbol{\theta}_c$. Here, $\mathbf{K}_{\text{cen}} \in \mathbb{R}^{k \times k}, \mathbf{K}_{\text{off}} \in \mathbb{C}^{k \times k}$ are trainable parameter matrices.

**Free parameter**: A strength of the geometric operator is that it can be applied compositionally, whereas a weakness is that its expressive power may be limited, since the two $\mathbf{K}$ matrices apply on *all* occurrences of Kleene plus. As an alternative, for every relation $r$, we define free parameter rotation box embeddings of $r^+$ denoted by $\mathbf{r}^+$. The query $q = (e_1, r^+, ?)$ is then represented as $(\mathbf{e}_1 \odot \text{Cen}(\mathbf{r}^+), \text{Off}(\mathbf{r}^+))$. Such a formulation cannot handle compositional queries like $(e_1, (r_1 \vee r_2)^+, ?)$ or $(e_1, (r_1/r_2)^+, ?)$. However, the motivation behind this formulation is that most of the Kleene plus queries in user query logs of Wikidata do not require Kleene plus operator to be compositional (Table 9).

### 4.3 Disjunction

We follow two approaches for modeling disjunction ($\vee$). In one case, we split a disjunction into independent non-disjunctive regexes. In the second case, we train a disjunction operator.

**Aggregation:** This approach follows Ren et al. [2020], who proposed transforming EPFO queries to Disjunctive Normal Form (DNF), moving disjunction operator ($\vee$) to the last step of computation graph. While all regex expressions cannot be transformed to a DNF form, a significant fraction (75%) of regex queries in our dataset formed from Wikidata user query logs (Section 5) can be expressed as a disjunction of a tractable number of non-disjunctive constituent queries $q = q_1 \vee q_2 \vee \ldots \vee q_N$. For example, $q = (e_1, r_1/(r_2^+ \vee r_3^+), ?)$ can be expressed as $q_1 \vee q_2$ where $q_1 = (e_1, r_1/r_2^+, ?)$ and $q_2 = (e_1, r_1/r_3^+, ?)$. For such queries, following Ren et al. [2020], we compute scores independently for each query, and aggregate them based on the minimum distance to the closest query box, i.e. $\text{dist}(\mathbf{e}; \mathbf{q}) = \text{Min}(\{\text{dist}(\mathbf{e}; \mathbf{q}_1), \text{dist}(\mathbf{e}; \mathbf{q}_2), \ldots, \text{dist}(\mathbf{e}; \mathbf{q}_N)\})$. Notice that this approach is not applicable if such a query decomposition is not feasible, such as for $(r_1 \vee r_2)^+$.

**DeepSets:** Zaheer et al. [2017] introduce a general architecture called DeepSets for functions where the inputs are permutation invariant sets. As disjunction ($\vee$) is permutation invariant, we use DeepSets to model it. We define an operator $\mathcal{D}$ which takes the rotation box embeddings $\mathbf{c}_1, \mathbf{c}_2, \ldots, \mathbf{c}_N$ of regular expressions $c_1, c_2, \ldots, c_N$ (corresponding to rotation angles $\boldsymbol{\theta}_{c_1}, \boldsymbol{\theta}_{c_2}, \ldots, \boldsymbol{\theta}_{c_N}$), and returns the embedding of $c = c_1 \vee c_2 \vee \ldots \vee c_N$, denoted by $\mathbf{c}$. *RotatE-Box* defines $\mathcal{D}(\mathbf{c}_1, \mathbf{c}_2, \ldots, \mathbf{c}_N) = (e^{i\boldsymbol{\theta}_c}, \text{Off}(\mathbf{c}))$ as follows:

$$\boldsymbol{\theta}_c = \mathbf{W}_{\text{cen}} \cdot \Psi(\text{MLP}_{\text{cen}}(\boldsymbol{\theta}_{c_1}), \text{MLP}_{\text{cen}}(\boldsymbol{\theta}_{c_2}), \ldots, \text{MLP}_{\text{cen}}(\boldsymbol{\theta}_{c_N}))$$
$$\text{Off}(\mathbf{c}) = \mathbf{W}_{\text{off}} \cdot \Psi(\text{MLP}_{\text{off}}(\text{Off}(\mathbf{c}_1)), \text{MLP}_{\text{off}}(\text{Off}(\mathbf{c}_2)), \ldots, \text{MLP}_{\text{off}}(\text{Off}(\mathbf{c}_N)))$$

where $\text{MLP}_{\text{cen}}, \text{MLP}_{\text{off}}$ are Multi-Layer Perceptron networks, $\Psi$ is a permutation-invariant vector function (element-wise min, max or mean), and $\mathbf{W}_{\text{cen}}, \mathbf{W}_{\text{off}}$ are trainable matrices.

### 4.4 Training objective

Following the previous work [Sun et al., 2019, Ren et al., 2020], we optimize a negative sampling loss to train RotatE-Box. Let $\gamma$ be a constant scalar margin. For a query $q$ with

answer entity $e$, and a *negative entity* not in answer set $e'_i$, the loss is defined as:

$$L = -\log\sigma(\gamma - \text{dist}(\mathbf{e}; \mathbf{q})) - \sum_{i=1}^{n} \frac{1}{n}\log\sigma(\text{dist}(\mathbf{e}'_i, \mathbf{q}) - \gamma) \tag{4}$$

## 5. Dataset

We contribute two new regex query datasets for our task – the first is based on Wikidata[4] [Vrandečić and Krötzsch, 2014a], a large popular open-access knowledge base, and the second is based on FB15K [Bordes et al., 2013], a subset of the Freebase KB commonly used in KBC tasks. For both datasets, we first split facts in original KB $\mathcal{G}$ into three subsets to construct $\mathcal{G}_{train}$, $\mathcal{G}_{dev}$, and $\mathcal{G}_{test}$ (using existing splits for FB15K). These are used to split regex queries into train, dev and test sets.

**Wiki100-Regex:** The Wikidata KB contains over 69 million entities, which makes it difficult to experiment with directly. We first construct a dense subset based on English Wikidata, with 100 relations and 41,291 entities, and name it *Wiki100*. This knowledge base contains 443,904 triples (Appendix B.1 has more details on Wiki100 construction).

Existing work on KBC with complex queries uses datasets built using random walks over KB [Guu et al., 2015, Kotnis et al., 2021] or uniform sampling over query DAG structure [Hamilton et al., 2018, Ren et al., 2020]. The availability of a large-volume of query (SPARQL) logs over Wikidata, offers us a unique opportunity to use real-world user queries for Wiki100-Regex. On studying the query logs, we find that 99% of regex queries (excluding single-hop queries) are of five types – $(e_1, r_1^+, ?)$, $(e_1, r_1^+/r_2^+, ?)$, $(e_1, r_1/r_2^+, ?)$, $(e_1, r_1 \vee r_2, ?)$, and $(e_1, (r_1 \vee r_2)^+, ?)$. We retain all regex queries from these types as the final set of queries.

To split the user queries into train/dev/test, we traverse each user query $(e_1, c, ?)$ up to depth 5, where $c$ is the regex expression. For every answer entity $e_2$ reachable from $e_1$ following $c$ in $\mathcal{G}_{train}$, we place $(e_1, c, e_2)$ in train split. If $e_2$ is not reachable in $\mathcal{G}_{train}$, but reachable in $\mathcal{G}_{train} \cup \mathcal{G}_{dev}$, we place $(e_1, c, e_2)$ in dev split, else in test split. We further augment the training split with random walks on $\mathcal{G}_{train}$ for these 5 query types. The final regex query type distribution is given in Table 9 in Appendix B.2.

**FB15K-Regex:** This dataset is constructed with the goal of testing our models on a higher diversity of regex queries than Wiki100-Regex. We identify 21 different types of regex expressions, using up to three different FB15K relations along with Kleene plus (+) and/or disjunction ($\vee$) operators. For each regex expression type $c$, we enumerate a set of compatible relations paths

| Dataset | Train | Dev | Test |
|---|---|---|---|
| **FB15K** | 483,142 | 50,000 | 59,071 |
| **Wiki100** | 389,795 | 21,655 | 21,656 |
| **FB15K-Regex** | 580,085 | 112,491 | 200,237 |
| **Wiki100-Regex** | 1,205,046 | 66,131 | 62,818 |

Table 4: Query distribution of the base KBs and corresponding regex datasets.

$l(c)$, up to length 5. For a random source entity $e_1$, the answer set of regex query $q = (e_1, c, ?)$ is formed by aggregating the answer sets of compatible relations path queries, i.e. $[\![q]\!] = \bigcup_{p \in l(c), len(p) <= 5} [\![(e_1, p, ?)]\!]$. We discard generic queries that have answer set size

---

[4] https://www.wikidata.org/

of over 50 entities. This procedure results in a highly skewed dataset, with 3 query types accounting for more than 80% of the regex queries. We reduce the skew in the query distribution by undersampling frequent query types. The final dataset statistics for each regex query type are summarized in Table 8 in Appendix B.2. The split into train/dev/test sets follows a procedure similar to one used when constructing Wiki100-Regex data. The dataset statistics of both the datasets are summarized in Table 4.

## 6. Experiments

In this section, we run experiments on the two datasets introduced in Section 5. We compare our RotatE-Box model with two competitive methods for complex query answering – Query2Box and BetaE, along with a close baseline, RotatE. We also wish to find out the performance difference between compositional and non-compositional variants to handle disjunction and Kleene plus.

### 6.1 Experimental Setup

**Baselines and model variants:** We note that regex variants of distribution-based models like BetaE and distance-based models like Query2Box and RotatE can also be created by applying similar ideas as in Sections 4.1-4.3 (Appendix A.2). For each of the three models, we experiment with all possible combination of operators:

- *KBC + Aggregation (BASELINE):* In this variant, we only train on single hop queries (link prediction), $(e_1, r, ?)$, and evaluate the model on regex queries by treating $r^+$ as $r$, and using Aggregation operator for handling disjunction.
- *Free parameter + Aggregation:* In this variant, Kleene plus is modeled as a free parameter for every relation and disjunction is handled via Aggregation.
- *Free parameter + DeepSets:* In this variant, Kleene plus is modeled as a free parameter and disjunction is handled via DeepSets.
- *Projection + Aggregation:* In this variant, we use Projection for Kleene plus and aggregation for disjunction.
- *Projection + DeepSets (COMP):* In this variant, we use Projection for Kleene plus and DeepSets for disjunction. Unlike other variants, both the operations are compositional and hence, can be applied to any arbitrary regex query.

Apart from *Projection + DeepSets (COMP)*, none of the other variants listed above can answer every query type, due to the presence of non-compositional operators for regex. For example, the query type $(e_1, (r_1/r_2)^+, ?)$ cannot be answered by these variants. In such cases, the query is not evaluated and the rank of correct entity is simply returned as $\infty$.

**Evaluation metrics:** Given a regex query $q = (e_1, c, ?)$, we evaluate validity of $e_2$ as an answer of the query by ranking all entities, while filtering out other entities from the answer set $[\![q]\!]$ [Bordes et al., 2013]. The candidate set for query $q$, $\mathcal{C}(q)$, can then be defined as $\mathcal{C}(q) = \{e_2\} \cup \{\mathcal{E} - [\![q]\!]\}$. We calculate the rank of $e_2$ amongst $\mathcal{C}(q)$ and compute ranking metrics such as Mean Reciprocal Rank (MRR) and Hits at $K$ (HITS@K).

**Implementation details:** An embedding in complex vector space has two trainable parameters per dimension (real and imaginary part) whereas a real valued embedding has

| Model | FB15K-Regex | | | | Wiki100-Regex | | | |
|---|---|---|---|---|---|---|---|---|
| | MRR | HITS@1 | HITS@5 | HITS@10 | MRR | HITS@1 | HITS@5 | HITS@10 |
| Query2Box (BASELINE) | 14.19 | 7.91 | 19.96 | 24.92 | 5.69 | 1.22 | 8.63 | 15.22 |
| Query2Box (Free parameter + Aggregation) | 21.92 | 12.54 | 31.09 | 39.45 | 28.65 | 12.32 | 47.84 | 54.50 |
| Query2Box (Free parameter + DeepSets) | 22.23 | 13.01 | 31.26 | 39.84 | 29.07 | 13.17 | 47.69 | 54.50 |
| Query2Box (Projection + Aggregation) | 21.74 | 12.42 | 30.84 | 39.28 | 29.42 | 13.74 | 47.94 | 54.45 |
| Query2Box (COMP) | 24.05 | 13.91 | 32.19 | 40.67 | 37.12 | 16.14 | 61.24 | 70.07 |
| BetaE (BASELINE) | 3.84 | 2.00 | 5.33 | 7.17 | 7.16 | 2.26 | 11.65 | 18.92 |
| BetaE (Free parameter + Aggregation) | 23.36 | 15.74 | 30.44 | 38.97 | 31.00 | 23.76 | 39.12 | 45.00 |
| BetaE (Free parameter + DeepSets) | 23.51 | 15.67 | 30.82 | 39.15 | 30.63 | 23.50 | 38.42 | 44.51 |
| BetaE (Projection + Aggregation) | 23.32 | 15.62 | 30.53 | 38.99 | 31.22 | 23.91 | 39.07 | 45.60 |
| BetaE (COMP) | 25.90 | 17.66 | 33.59 | 42.26 | 41.93 | 33.22 | 51.45 | 58.83 |
| RotatE (BASELINE) | 11.47 | 7.46 | 14.68 | 18.95 | 3.67 | 1.31 | 4.65 | 7.99 |
| RotatE (Free parameter + Aggregation) | 20.63 | 13.18 | 27.47 | 34.99 | 36.36 | 29.41 | 44.10 | 49.78 |
| RotatE (Free parameter + DeepSets) | 21.23 | 13.64 | 28.15 | 35.77 | 36.07 | 27.45 | 46.05 | 51.85 |
| RotatE (Projection + Aggregation) | 20.52 | 12.98 | 27.34 | 34.90 | 33.93 | 22.25 | 47.69 | 53.70 |
| RotatE (COMP) | 22.87 | 14.60 | 30.45 | 38.41 | 45.58 | 33.26 | 59.84 | 67.67 |
| RotatE-Box (BASELINE) | 10.73 | 7.23 | 13.82 | 17.08 | 4.63 | 1.92 | 6.16 | 10.36 |
| RotatE-Box (Free parameter + Aggregation) | 24.11 | 16.13 | 31.53 | 39.74 | 39.29 | 30.25 | 50.00 | 55.33 |
| RotatE-Box (Free parameter + DeepSets) | 24.15 | 15.96 | 31.93 | 40.19 | 39.98 | 31.54 | 50.09 | 55.33 |
| RotatE-Box (Projection + Aggregation) | 23.82 | 15.70 | 31.50 | 39.63 | 36.74 | 27.14 | 47.97 | 53.76 |
| RotatE-Box (COMP) | **26.55** | **17.80** | **34.90** | **43.56** | **49.29** | **37.35** | **63.56** | **70.95** |

Table 5: Overall performance of all model variants over two benchmark datasets

only one. Therefore, to keep the number of trainable parameters comparable, we use $k = 400$ for complex embedding based models (RotatE and RotatE-Box) and $k = 800$ for real embedding based models (Query2Box). Relation and entity embeddings of all the models are first trained on single-hop queries $(e_1, r, ?)$, and then optimized for regex queries along with other trainable parameters of regex operators. For Kleene plus operator (free parameter variant), $\mathbf{r}^+$ is initialized as $\mathbf{r}$. We use element-wise minimum function for $\Psi$. Other training specific details and hyperparameter selection are in Appendix C.1.

### 6.2 Results and Discussions

Table 5 shows the main results of different embedding based models with different variants for Kleene plus and disjunction on both datasets. Overall, we observe that the compositional variants of all models outperform non-compositional ones. RotatE-Box obtains significant improvements over other distance-based methods – RotatE and Query2Box (correspondingly for all variants). RotatE-Box variants also perform comparably to BetaE models for FB15K-Regex, and significantly outperform them for Wiki100-Regex. Comparing best and second best results, RotatE-Box(COMP) achieves 0.65 points MRR over BetaE(COMP) for FB15K-Regex, and over 3.7 points MRR over RotatE(COMP) for Wiki100-Regex. The massive improvement of both COMP and other variants over BASELINE highlights the necessity of training KBC models beyond single-hop queries.

Recall that for each model, all variants except COMP contain only non-compositional regex operators, hence, a portion of queries are not evaluated, and the score for those queries is returned as 0. To get better insights into operators, we look at the subset of regex queries that are answerable by all variants. In our experiments, this subset includes all regex query types except $(e_1, (r_1 \vee r_2)^+, ?)$. The results of this ablation study are presented in Table 6. While RotatE-Box maintains its superior performance over other models on

| Model | FB15K-Regex | | | | Wiki100-Regex | | | |
|---|---|---|---|---|---|---|---|---|
| | MRR | HITS@1 | HITS@5 | HITS@10 | MRR | HITS@1 | HITS@5 | HITS@10 |
| Query2Box (Free parameter + Aggregation) | 23.12 | 13.23 | 32.80 | 41.61 | 37.89 | 16.30 | 63.28 | 72.09 |
| Query2Box (Free parameter + DeepSets) | 23.45 | 13.72 | 32.97 | 42.03 | 38.44 | 17.43 | 63.08 | 72.09 |
| Query2Box (Projection + Aggregation) | 22.93 | 13.10 | 32.54 | 41.43 | 38.92 | 18.17 | 63.42 | 72.02 |
| Query2Box (COMP) | 23.29 | 13.59 | 32.69 | 41.73 | 40.38 | 20.63 | 63.43 | 72.27 |
| BetaE (Free parameter + Aggregation) | 24.65 | 16.60 | 32.11 | 41.11 | 41.00 | 31.43 | 51.74 | 59.52 |
| BetaE (Free parameter + DeepSets) | 24.80 | 16.53 | 32.51 | 41.29 | 40.52 | 31.08 | 50.82 | 58.87 |
| BetaE (Projection + Aggregation) | 24.60 | 16.48 | 32.21 | 41.13 | 41.30 | 31.63 | 51.68 | 60.32 |
| BetaE (COMP) | 24.89 | 16.65 | 32.56 | 41.30 | 43.52 | 34.56 | 53.35 | 61.04 |
| RotatE (Free parameter + Aggregation) | 21.76 | 13.90 | 28.98 | 36.91 | 48.09 | 38.90 | 58.33 | 65.85 |
| RotatE (Free parameter + DeepSets) | 22.39 | 14.38 | 29.69 | 37.73 | 47.71 | 36.31 | 60.92 | 68.59 |
| RotatE (Projection + Aggregation) | 21.64 | 13.69 | 28.84 | 36.81 | 44.89 | 29.43 | 63.08 | 71.03 |
| RotatE (COMP) | 21.97 | 13.89 | 29.30 | 37.31 | 47.45 | 35.05 | 61.94 | 69.96 |
| RotatE-Box (Free parameter + Aggregation) | 25.43 | **17.01** | 33.26 | 41.92 | 51.97 | 40.01 | 66.14 | **73.19** |
| RotatE-Box (Free parameter + DeepSets) | **25.48** | 16.83 | **33.68** | **42.39** | **52.89** | **41.73** | **66.26** | **73.19** |
| RotatE-Box (Projection + Aggregation) | 25.13 | 16.56 | 33.23 | 41.80 | 48.61 | 35.91 | 63.46 | 71.11 |
| RotatE-Box (COMP) | 25.29 | 16.58 | 33.56 | 42.32 | 51.51 | 39.75 | 65.82 | 73.10 |

Table 6: Performance on subset of regex query types answerable by all variants. Best overall score is in bold. Best score amongst variants of the same model is underlined.

| Model | FB15K | | | | Wiki100 | | | |
|---|---|---|---|---|---|---|---|---|
| | MRR | HITS@1 | HITS@5 | HITS@10 | MRR | HITS@1 | HITS@5 | HITS@10 |
| Query2Box (BASELINE) | 67.86 | 56.39 | 80.73 | 84.73 | 22.44 | 0.78 | 47.56 | 57.12 |
| Query2Box (Free parameter + Aggregation) | 43.92 | 30.15 | 59.55 | 67.56 | 21.00 | 0.59 | 44.80 | 55.11 |
| Query2Box (COMP) | 42.49 | 29.18 | 57.37 | 65.76 | 20.74 | 0.59 | 44.17 | 54.87 |
| BetaE (BASELINE) | 50.63 | 40.56 | 61.99 | 69.09 | 27.60 | 16.88 | 39.43 | 46.98 |
| BetaE (Free parameter + Aggregation) | 42.79 | 32.21 | 54.40 | 62.44 | 19.98 | 8.23 | 25.31 | 34.25 |
| BetaE (COMP) | 42.02 | 31.59 | 53.60 | 61.70 | 19.91 | 9.51 | 30.74 | 40.87 |
| RotatE (BASELINE) | **76.11** | **68.77** | **84.91** | **88.47** | 38.74 | 28.54 | 50.30 | 57.62 |
| RotatE (Free parameter + Aggregation) | 36.93 | 26.06 | 48.82 | 57.08 | 29.97 | 17.48 | 43.86 | 52.29 |
| RotatE (COMP) | 34.95 | 24.24 | 46.42 | 55.02 | 23.72 | 6.74 | 43.10 | 53.16 |
| RotatE-Box (BASELINE) | 73.76 | 66.35 | 82.67 | 86.60 | **40.15** | **30.12** | **51.23** | 58.40 |
| RotatE-Box (Free parameter + Aggregation) | 44.53 | 33.11 | 56.97 | 65.24 | 30.67 | 13.72 | 49.52 | **58.77** |
| RotatE-Box (COMP) | 43.00 | 31.48 | 55.50 | 64.02 | 28.47 | 10.87 | 48.33 | 57.93 |

Table 7: Performance on KBC queries ($e_1$, r, ?) before and after training on regex queries.

this subset of regex queries, both the formulations of Kleene plus operator – Projection and Free parameter, have almost similar performance, with Free parameter variant showing marginally higher MRR and HITS@1 in most cases. In particular, (Free parameter + DeepSets) for RotatE-Box on Wiki100-Regex is better than (COMP) by more than 1.3 points MRR. This suggests that the simple Projection operator does not always achieve quite the same expressiveness as free parameters. We leave the study of deeper trainable projection operators for future work.

### 6.2.1 Impact of regex training on single-hop queries

The regex query training is initialized with entity and relation embeddings trained on single–hop queries. However, when the model is optimized on regex queries, the performance on

single–hop queries drops dramatically for both COMP and (Free parameter + Aggregation) variants. Similar trend is observed in other variants as well. The results are summarized in Table 7.

We believe this could the the result of entity embeddings being optimized for two conflicting objectives. Consider the following example. Let $q_1$ represent the query $(e_1, r, ?)$ and let $e_2$ be an answer to $q_1$. Similarly, let $q_2 = (e_2, r, ?)$ and $e_3$ be an answer for $q_2$. In theory, from KBC training we get $\mathbf{q_1} - \mathbf{e_2} = \mathbf{q_2} - \mathbf{e_3} = \mathbf{0}$, i.e. $\mathbf{e_3} - \mathbf{e_2} = \mathbf{q_2} - \mathbf{q_1} \neq \mathbf{0}$ for any distance-based model. However, for regex query $q' = (e_1, r^+, ?)$, for which $e_2$ and $e_3$ are both answers, $\mathbf{e_3} - \mathbf{e_2} = \mathbf{q'} - \mathbf{q'} = \mathbf{0}$. Hence, distance-based embedding models cannot simultaneously model both KBC and regex queries. This hypothesis can also be extended to distribution-based models like BetaE, which models distance between entity and query as KL divergence between their respective distributions. Similar phenomenon has been observed in previous works on complex queries [Ren et al., 2020]. We leave it to future work to optimize for complex queries without losing performance on single–hop queries. We also observe that on Wiki100, RotatE-Box has a higher performance than RotatE, suggesting that RotatE-Box may have some value for pure KBC tasks also.

## 7. Conclusion

In this work we present the novel task of answering regex queries over incomplete knowledge bases. We provide two datasets and query workloads for the task and provide baselines for training regex operators. Of the two datasets we present, Wiki100-Regex is a novel challenging benchmark based entirely on a real-world KB and query logs. We also provide a new model *RotatE-Box*, which models more relational inference patterns than other approaches. Our code and data is available at https://github.com/dair-iitd/kbi-regex.

While the baselines of this work are the first step in modeling regex operators, there is a lot of scope for future research. Particularly, Kleene Plus poses novel modeling challenges – it is an idempotent unary operator $((r^+)^+ = r^+)$ *and* an infinite union of path queries $(r^+ = r \vee (r/r) \vee (r/r/r) \ldots)$. In the future, we plan to work on constructing parameterized operator functions, which honor these properties of the operator, and can be applied compositionally and recursively, while being sufficiently expressive.

## Acknowledgments

This work was supported by an IBM AI Horizons Network (AIHN) grant, grants by Google, Bloomberg and 1MG, a Visvesvaraya faculty award by Govt. of India, and a Jai Gupta Chair fellowship to Mausam. Srikanta Bedathur was partially supported by a DS Chair of AI fellowship. We thank the IITD HPC facility for compute resources.

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

## Appendix A. Model

### A.1 Proof of Theorem 1

*Proof.* Consider a relation $r_1$ represented by $(\mathrm{Cen}(\mathbf{r}_1), \mathrm{Off}(\mathbf{r}_1)) \in \mathbb{C}^k$. The query representation of $q_1 = (e_1, r_1, ?)$ will be

$$\mathrm{Cen}(\mathbf{q}_1) = \mathrm{Cen}(\mathbf{e}_1) \odot \mathrm{Cen}(\mathbf{r}_1)$$
$$\mathrm{Off}(\mathbf{q}_1) = \mathrm{Off}(\mathbf{r}_1))$$

If an entity $e_2$ is in the answer set of $q_1$, then it must be inside the query box of $q_1$, i.e:

$$\mathrm{Re}(\mathrm{Cen}(\mathbf{q}_1) - \mathrm{Off}(\mathbf{q}_1)) \le \mathrm{Re}(\mathrm{Cen}(\mathbf{e}_2)) \le \mathrm{Re}(\mathrm{Cen}(\mathbf{q}_1) + \mathrm{Off}(\mathbf{q}_1))$$
$$\mathrm{Im}(\mathrm{Cen}(\mathbf{q}_1) - \mathrm{Off}(\mathbf{q}_1)) \le \mathrm{Im}(\mathrm{Cen}(\mathbf{e}_2)) \le \mathrm{Im}(\mathrm{Cen}(\mathbf{q}_1) + \mathrm{Off}(\mathbf{q}_1))$$

Now consider a relation $r_2$ such that $\mathrm{Cen}(\mathbf{r}_2) = \mathrm{Cen}(\mathbf{r}_1)$, $\mathrm{Re}(\mathrm{Off}(\mathbf{r}_2)) \ge \mathrm{Re}(\mathrm{Off}(\mathbf{r}_1))$ and $\mathrm{Im}(\mathrm{Off}(\mathbf{r}_2)) \ge \mathrm{Im}(\mathrm{Off}(\mathbf{r}_1))$. Hence, for query $q_2 = (e_1, r_2, ?)$ represented by:

$$\mathrm{Cen}(\mathbf{q}_2) = \mathrm{Cen}(\mathbf{e}_1) \odot \mathrm{Cen}(\mathbf{r}_2)$$
$$\mathrm{Off}(\mathbf{q}_2) = \mathrm{Off}(\mathbf{r}_2))$$

the following is true:

$$\mathrm{Re}(\mathrm{Cen}(\mathbf{q}_2) - \mathrm{Off}(\mathbf{q}_2)) \le \mathrm{Re}(\mathrm{Cen}(\mathbf{e}_2)) \le \mathrm{Re}(\mathrm{Cen}(\mathbf{q}_2) + \mathrm{Off}(\mathbf{q}_2))$$
$$\mathrm{Im}(\mathrm{Cen}(\mathbf{q}_2) - \mathrm{Off}(\mathbf{q}_2)) \le \mathrm{Im}(\mathrm{Cen}(\mathbf{e}_2)) \le \mathrm{Im}(\mathrm{Cen}(\mathbf{q}_2) + \mathrm{Off}(\mathbf{q}_2))$$

Hence, the entity $e_2$ is inside the query box of $q_2$. Therefore, $r_1(e_1, e_2)$ implies $r_2(e_1, e_2)$. Thus, *RotatE-Box* can model hierarchy. $\square$

### A.2 Regex Variants of Baseline Models

In this section we describe how the formulations of regex operators (Section 4.1- 4.3) apply to *Query2box* and *RotatE*. Similar ideas can be easily extended to BetaE as well.

**Query2Box**: Each entity $e$ is modeled as a point $\mathbf{e} \in \mathbb{R}^k$ and each relation $r$ is modeled as a box $\mathbf{r}$, represented as $(\mathrm{Cen}(\mathbf{r}), \mathrm{Off}(\mathbf{r})) \in \mathbb{R}^{2k}$. Unlike RotatE-Box, the modulus of $\mathrm{Cen}(\mathbf{r})$ is not constrained. The representation of a single-hop query $q = (e_1, r, ?)$ is constructed by *translating* the center and adding the offset, i.e. $\mathbf{q} = (\mathbf{e}_1 + \mathrm{Cen}(\mathbf{r}), \mathrm{Off}(\mathbf{r}))$. Generalizing this to path queries, the path embedding $\mathbf{p}_n$ for relation path $r_1/r_2/\ldots/r_n$ is computed as follows:

$$\mathrm{Cen}(\mathbf{p}_{i+1}) = \mathrm{Cen}(\mathbf{p}_i) + \mathrm{Cen}(\mathbf{r}_{i+1}) \qquad \mathrm{Off}(\mathbf{p}_{i+1}) = \mathrm{Off}(\mathbf{p}_i) + \mathrm{Off}(\mathbf{r}_{i+1}) \qquad (5)$$

Here, $\mathbf{p}_1 = \mathbf{r}_1$. Query $q = (e_1, r_1/r_2/\ldots/r_n, ?)$ is represented as $\mathbf{q} = (\mathbf{e}_1 + \mathrm{Cen}(\mathbf{p}_n), \mathrm{Off}(\mathbf{p}_n))$.

Modeling Kleene plus as Projection follows from Section 4.2, however, for Query2Box, the projection matrices are applied to actual embeddings, rather rather rotation angles. Given representation $\mathbf{c}$ for a regex expression $c$, $c^+$ is modeled as $kp(\mathbf{c}) = \mathbf{c}' = \mathbf{Kc}$, $\mathbf{K} \in \mathbb{R}^{k \times k}$ While modeling Kleene plus as a Free parameter, the query $q = (e_1, r^+, ?)$ is represented as $(\mathbf{e}_1 + \text{Cen}(\mathbf{r}^+), \text{Off}(\mathbf{r}^+))$.

We follow the original implementation of Query2Box [Ren et al., 2020] for modeling Disjunction as aggregation. When modeling disjunction with DeepSets, $c = c_1 \vee c_2 \vee \ldots \vee c_N$ is represented as:

$$\text{Cen}(\mathbf{c}) = \mathbf{W}_{\text{cen}} \cdot \Psi(\text{MLP}_{\text{cen}}(\text{Cen}(\mathbf{c}_1)), \text{MLP}_{\text{cen}}(\text{Cen}(\mathbf{c}_2)), \ldots, \text{MLP}_{\text{cen}}(\text{Cen}(\mathbf{c}_N)))$$
$$\text{Off}(\mathbf{c}) = \mathbf{W}_{\text{off}} \cdot \Psi(\text{MLP}_{\text{off}}(\text{Off}(\mathbf{c}_1)), \text{MLP}_{\text{off}}(\text{Off}(\mathbf{c}_2)), \ldots, \text{MLP}_{\text{off}}(\text{Off}(\mathbf{c}_N)))$$

**RotatE**: For RotatE, entities and relations are modeled as points in the complex space, i.e. $\mathbf{e} \in \mathbb{C}^k$ and $\mathbf{r} \in \mathbb{C}^k$. The modulus of $\mathbf{r}$ is constrained to be 1 in every dimension ($r = e^{i\boldsymbol{\theta}_r}$). The single-hop query $q = (e_1, r, ?)$ is constructed by *rotating* the entity embedding ($\mathbf{q} = \mathbf{e}_1 \odot \mathbf{r}$). The target entity is scored by its distance from the query embedding $\text{dist}(\mathbf{e}; \mathbf{q}) = |\mathbf{q} - \mathbf{e}|$. Computation of the relation path follows Section 4.1, but ignoring the offset term. Therefore, $q = (e_1, r_1/r_2/\ldots/r_n, ?)$ is represented as $\mathbf{q} = \mathbf{e}_1 \odot \mathbf{p}_n$, where $\mathbf{p}_n = \mathbf{r}_1 \odot \mathbf{r}_2 \odot \ldots \odot \mathbf{r}_n$.

Given representation $\mathbf{c}$ (corresponding to rotation by angle $\boldsymbol{\theta}_c$) for a regex expression $c$, $c^+$ is modeled as $kp(\mathbf{c}) = \mathbf{c}' = e^{i\boldsymbol{\theta}_{c'}}$, where $\boldsymbol{\theta}_{c'} = \mathbf{K}\boldsymbol{\theta}_c$, $\mathbf{K} \in \mathbb{R}^{k \times k}$, when using Projection for Kleene plus. The free parameter embeddings of $r^+$ is denoted by $\mathbf{r}^+$. For RotatE, The query $q = (e_1, r^+, ?)$ is then represented as $\mathbf{q} = \mathbf{e}_1 \odot \mathbf{r}^+$.

Modeling disjunction as aggregation is similar to that described in Section 4.3, using $\text{dist}(\mathbf{e}; \mathbf{q})$ as defined above. For DeepSets, given the embeddings $\mathbf{c}_1, \mathbf{c}_2, \ldots, \mathbf{c}_N$ (corresponding to rotation angles $\boldsymbol{\theta}_{c_1}, \boldsymbol{\theta}_{c_2}, \ldots, \boldsymbol{\theta}_{c_N}$) of regular expressions $c_1, c_2, \ldots, c_N$, $c = c_1 \vee c_2 \vee \ldots \vee c_N$ is represented as:

$$\mathbf{c} = e^{i\boldsymbol{\theta}_c}, \qquad \boldsymbol{\theta}_c = \mathbf{W} \cdot \Psi(\text{MLP}(\boldsymbol{\theta}_{c_1}), \text{MLP}(\boldsymbol{\theta}_{c_2}), \ldots, \text{MLP}(\boldsymbol{\theta}_{c_N}))$$

where MLP is a Multi-Layer Perceptron network, $\Psi$ is a permutation-invariant vector function, and $\mathbf{W}$ is a trainable matrix.

## Appendix B. Dataset

### B.1 Wiki100 Knowledge Base

Wikidata is a collaboratively edited multilingual knowledge base that manages factual data in Wikipedia [Vrandečić and Krötzsch, 2014b]. It contains more than 69 million entities which makes it difficult to experiment with directly. We construct a dense subset based on English Wikidata as follows: we first selected entities and relations which have at least 100 instances in Wikidata and selected triples mentioning only the top-100 frequent relations. The resulting dataset had highly skewed distribution of relations with 6 relations accounting for more than 95% of all triples. In order to smooth this distribution and to include a significant number of all relations, we undersample facts containing these 6 relations, and oversample others. Finally, we recursively filtered sparse entities mentioned less than $n$ times (decreasing $n$ from 10 to 1) to obtain a knowledge base which we call as **Wiki100**, with $443,904$ triples over $41,291$ entities and 100 unique relations, which we randomly split into train/dev/test.

**B.2 Regex Query Distribution**

| Query type | Train | Valid | Test |
|---|---|---|---|
| $(e_1, r_1^+, ?)$ | 24,476 | 4,614 | 8,405 |
| $(e_1, r_1/r_2, ?)$ | 25,378 | 4,927 | 8,844 |
| $(e_1, r_1^+/r_2^+, ?)$ | 26,391 | 4,978 | 9,028 |
| $(e_1, r_1^+/r_2^+/r_3^+, ?)$ | 25,470 | 4,878 | 8,816 |
| $(e_1, r_1/r_2^+, ?)$ | 26,335 | 5,007 | 9,062 |
| $(e_1, r_1^+/r_2, ?)$ | 27,614 | 5,229 | 9,429 |
| $(e_1, r_1^+/r_2^+/r_3, ?)$ | 27,865 | 5,283 | 9,509 |
| $(e_1, r_1^+/r_2/r_3^+, ?)$ | 26,366 | 5,058 | 9,159 |
| $(e_1, r_1/r_2^+/r_3^+, ?)$ | 26,366 | 5,045 | 9,099 |
| $(e_1, r_1/r_2/r_3^+, ?)$ | 26,703 | 5,155 | 9,313 |
| $(e_1, r_1/r_2^+/r_3, ?)$ | 28,005 | 5,380 | 9,688 |
| $(e_1, r_1^+/r_2/r_3, ?)$ | 27,884 | 5,338 | 9,632 |
| $(e_1, r_1 \vee r_2, ?)$ | 30,080 | 5,828 | 9,664 |
| $(e_1, (r_1 \vee r_2)/r_3, ?)$ | 31,559 | 6,606 | 10,974 |
| $(e_1, r_1/(r_2 \vee r_3), ?)$ | 41,886 | 7,755 | 13,611 |
| $(e_1, r_1^+ \vee r_2^+, ?)$ | 23,109 | 4,469 | 8,367 |
| $(e_1, (r_1 \vee r_2)/r_3^+, ?)$ | 27,658 | 5,738 | 9,711 |
| $(e_1, (r_1^+ \vee r_2^+)/r_3, ?)$ | 24,462 | 4,865 | 8,863 |
| $(e_1, r_1^+/(r_2 \vee r_3), ?)$ | 27,676 | 5,340 | 9,267 |
| $(e_1, r_1/(r_2^+ \vee r_3^+), ?)$ | 28,542 | 5,475 | 9,436 |
| $(e_1, (r_1 \vee r_2)^+, ?)$ | 26,260 | 5,523 | 10,360 |
| **Total** | 580,085 | 112,491 | 200,237 |

Table 8: Distribution of regex query types for FB15K-Regex dataset.

| Query type | Train | Valid | Test |
|---|---|---|---|
| $(e_1, r_1^+, ?)$ | 490,562 | 24,878 | 23,443 |
| $(e_1, r_1^+/r_2^+, ?)$ | 6,945 | 620 | 772 |
| $(e_1, r_1/r_2^+, ?)$ | 85,253 | 10,013 | 8,377 |
| $(e_1, r_1 \vee r_2, ?)$ | 274,012 | 14,900 | 14,915 |
| $(e_1, (r_1 \vee r_2)^+, ?)$ | 348,274 | 15,720 | 15,311 |
| **Total** | 1,205,046 | 66,131 | 62,818 |

Table 9: Distribution of regex query types for Wiki100-Regex dataset.

## Appendix C. Experiments

All the models are implemented in Pytorch-Lightning framework. We used NVIDIA V100 (16GB) GPUs from training the models. For all models, we use 2 GPUs when training on FB15K-Regex, and 4 GPUs when training on Wiki100-Regex.

### C.1 Hyperparameters

Model-specific hyperparameters such as margin $\gamma$ and down-weighting constant $\alpha$ are first tuned on single-hop query pre-training. Apart from learning rate $lr$, the same hyperparameters are used in regex query training. The ranges of hyperparameter grid search used for all our models are: $\gamma \in \{4.0, 8.0, 12.0, \ldots, 28.0, 32.0\}$, $\alpha \in$

| Hyperparameter | FB15K | Wiki100 |
|---|---|---|
| $\gamma$ | 24.0 | 20.0 |
| $\alpha$ | 0.2 | 0.2 |
| $lr$ | $10^{-4}$ | $10^{-3}$ |

Table 10: Hyperparameters used in training models on single-hop queries.

$\{0.2, 0.4, 0.6, 0.8, 1.0\}$, learning rate $lr = \{10^{-1}, 10^{-2}, 10^{-3}, 10^{-4}, 10^{-5}\}$. For a particular dataset, we found the same set of hyperparameters to perform best for all three models. The final values are reported in Table 10. All the models are optimized using Adam optimizer [Kingma and Ba, 2015]. For FB15K, learning rate is set to $10^{-4}$ for single hop as well as regex queries. For Wiki100, it is $10^{-3}$ when training on single hop queries and $10^{-4}$ when training on regex queries. We use a batch size of 1024 and sample 256 negative entities for each training query. The negative entities $e'_i$ are sampled uniformly for regex query training and adversarially while training RotatE and RotatE-Box on single–hop queries (as per [Sun et al., 2019]). For both datasets across all models, we train for 1000 epochs for single-hop queries, and 500 for regex queries, using early stopping on dev split MRR.