# OpenReview forum: "Regex Queries over Incomplete Knowledge Bases"
_AKBC.ws/2021/Conference — AKBC 2021_

### Official Review · Reviewer_RJB4 · 2021-07-21
**satisfied with the contributions; accept**

**Rating:** 7
**Confidence:** 4

**Review:**

**Summary**: This work is on the topic of answering complex queries over incomplete Knowledge Bases (KBs). A KB is a set of triples/edges of the form (e1, r, e2) where ei's are entities/nodes and r is a relation/edge-label. Given a start entity e1 & an ordered sequence of relations r1 / r2 / ....rk one has to return all entities at the end of the paths consistent with the relational path r1 / r2 / ....rk. Importantly, a fraction of the edges are removed during training. Authors consider a novel problem in which one is allowed to represent *multiple* relational paths using a *single* regular-expression possibly containing disjunction ($\lor$) and Kleene plus (+) operators. For a regular-expression c, c+ denotes the infinite set of expressions {c, c/c, c/c/c ...}.

To handle such powerful operators, authors devise a model RotatE-Box by generalizing 2 popular KB-embedding based approaches RotateE and Query2Box.
Authors construct 2 new datasets containing regular-expression queries: (a) Wiki100-regex on subset of Wikidata & (b) FB15K-regex using a subset of Freebase. They show that the new model outperforms the above 2 constituent approaches on these tasks.

**Strengths**:
1. Authors propose an interesting problem of answering regular-expression queries over KBs by including the Kleene plus operator. This operator is not straightforward to model via an embedding-based approach and can potentially lead to further research and novel modeling techniques.
2. Authors introduce 2 new datasets containing regular expression queries that again invites further research. During constructing, authors incorporate the statistics of regular-expression queries from actual users to up-sample the ones more often used in real-world.
3. Authors propose a generalization of past embedding-based approaches and show that the new model leads to improvements on the their datasets. Multiple modeling choices for the operators are explored.
4. Paper is written well and the experiments seem reproducible.

**Weaknesses**:
1. no major weaknesses

---

> ### Author Response · Authors · 2021-07-31
> **Thank you for constructive review and thoughtful comments**
>
> We thank the reviewer for the constructive review and thoughtful comments. We are glad the reviewer likes our work and acknowledges the challenges posed by our proposed task. We hope that our work would encourage further research in this direction. Thank you.

---

### Official Review · Reviewer_W8qj · 2021-07-23
**Interesting task and approach**

**Rating:** 7
**Confidence:** 3

**Review:**


The paper introduces a new task of solving queries consisting of disjunction and Kleene plus operators. They evaluate a previous model query2box on this task as well as propose a new model that combines query2box with a popular complex embedding model, RotatE. They show that the combination, termed RotatE-Box, is superior to using just RotatE or just Query2Box.

Pros
- The paper proposes both a new task, that seems relevant for AKBC, as well as a new dataset for the task (one constructed from existing FB15k and one new from wikidata)
- They propose a new method that combines two previous methods by extending quer2box to the complex space
- The paper is well written.
- Experiments are thorough and convincing that the approach has merit. I liked that the variations proposed for dealing with regex queries, like aggregation/deepsets etc, are also applied on the baselines for a fair evaluation.
- The authors provide the code and promise to make both code and data for the new tasks publicly available

Cons
- The regex query training seems to make the performance of single-hop KB completion worse. It’s not clear why this should happen and there is no good explanation provided.
- Since the work seems to be the first in proposing to use regex queries, it would have been interesting to also compare other embedding models such as ComplEx and DistMult on the set of query types they can answer.

---

> ### Author Response · Authors · 2021-07-31
> **Response to Reviewer W8qj**
>
> We thank the reviewer for the detailed review and for appreciating the novelty of our task. Our responses to the reviewer’s concerns are below.
>
>
> * *Regex query training seems to make the performance of single-hop KB completion worse*
>
>     It is true that this observation is dissatisfying and requires further inspection. We believe that this phenomenon occurs because KBC training and regex training optimize the entity embeddings for slightly contrasting objectives. For example, consider *(e1, r, e2)* and *(e2, r, e3)* to be two facts in the KB. Consider a simple distance-based method - TransE (similar observation holds for Q2B, RotatE, RotatE-box). Then, in theory, **e1 + r = e2** and **e2 + r = e3**, hence, **e2 - e1 = e3 - e2 = r**. Now consider *(e1, r+, ?)*. Both *e2* and *e3* are answers to this query. Hence on optimizing for regex queries, we get **e2 = e3 = e1 + r+**. Hence KBC training is trying to push **e2** and **e3** apart by **r** and regex training is trying to bring them close together. We have added this hypothesis to Section 6.2.1 (in red).
>
>     Having a box representation rather than a point possibly dilutes this effect as e2 and e3 need not be the same point as long as they are within the query box. This is perhaps why the drop in performance of single-hop KB completion is lesser for box methods like Query2Box and RotatE-box, as compared to point-based methods like RotatE.
> As we mention in the Conclusion (Section 7), we need models which overcome the modeling challenges of Kleene plus operator. Models that are successful in doing so might also be able to satisfactorily optimize for both single-hop KBC and regex queries.
>
>
> * *It would have been interesting to also compare other embedding models such as ComplEx and DistMult on the set of query types they can answer.*
>
>     In our early experiments, we experimented with ComplEx and DistMult and found them to perform very poorly on regex queries. We believe that this is probably due to the fact that similarity-based methods like ComplEx and DistMult cannot model composition relation pattern, which we believe is an essential property to model queries beyond single-hop queries. Hence, we build our model (RotatE-box) on top of distance-based methods like TransE, Query2Box, and RotatE, all of which can model composition.

---

### Official Review · Reviewer_szmE · 2021-07-23
**Incremental improvements using RotatE-Box model**

**Rating:** 6
**Confidence:** 4

**Review:**

**Paper summary:**
This paper combined two knowledge base completion models, RotatE and the Query2Box model, to form the proposed model RotatE-Box model. The proposed model is evaluated on a new task, knowledge base completion on regex queries. The authors also provided two new evaluation datasets for the regex queries. The proposed model achieved better performance on the regex query task compared to the two base models. But they did not compare it with enough baselines such as Beta Embeddings, and so on. And the proposed model performed much worse than the original model in the original knowledge base completion task.


**Strength**
- This paper is well-written, well-explained, and well-motivated. The motivation of having a RotateE-Box model is very clear since we want to model hierarchical relations in the rotated complex space, which should be much more flexible than Euclidean space.
- The ablation study in table 6 provides an insightful comparison between the simple projection model and free parameters, indicating free parameters sometimes provide more flexibility in learning and leads to better performance as well compared to simple project models.


**Weakness / Questions / Suggestions**:
- Several strong baselines are missing, including beta embeddings paper from the same group as Query2Box.
- The model’s performance on the single basic query is very limited compared to baseline models. Furthermore, it remains unclear about the real usage of the regex query.
- If the main difference between of RotateE-Box compared to RotatE is the hierarchical relation, then does the proposed dataset contain a lot of hierarchical relations? Why would the proposed model perform better than RotatE by a lot?
- Can we get the distribution of different types of regex queries in the new two datasets?

---

> ### Author Response · Authors · 2021-07-31
> **Response to Reviewer szmE**
>
> We thank the reviewer for the detailed review. Our responses to the reviewer’s concerns are below.
>
> * *Several strong baselines are missing, including beta embeddings paper from the same group as Query2Box.*
>
>     Since RotatE-Box is built on top of RotatE and Query2Box, our original experiments compared against those, to verify that the new model indeed provides benefit on top of its constituents.    We didn’t include BetaE in our experiments as it is a very different (distribution-based) model, whereas we were building our approach on top of distance-based models. On your suggestion, we ran BetaE on our regex queries. We have included the results in the Experiments section (Section 6) for FB15K-Regex. Indeed, BetaE performs better than Query2Box and RotatE and is close to but worse than RotatE-Box. RotatE-Box(COMP) performs better than BetaE(COMP) in 3 out of 4 metrics for FB15K-Regex. Due to time constraints, we could only run the experiments on 4 out of 5 baselines of FB15K-Regex. We will run BetaE for all baselines for both datasets and update the results in the final version of the paper.
>
>     While the comparison strengthens the experiments, we wish emphasize that the goal of our paper is to show that our contribution RotatE-Box performs better than the models it is based on - RotatE and Query2Box. We show this based on both theoretical properties and empirical results. The performance of BetaE opens another interesting direction: can we combine insights from distribution-based modeling and box-based modeling for possibly an even stronger performance on our task.
>
>     We would also like to reiterate that our contributions go beyond just the model. We introduce the task of regex question-answering, curate datasets, and provide baselines. We hope that our work would encourage further research in this direction.
>
>
> * *the proposed model performed much worse than the original model in the original knowledge base completion task*
>
>     Yes, we agree that the weaker performance on KBC queries is dissatisfying. Please see the response to reviewer W8qj (Con #1) and additional text in Section 6.2.1 (in red).
> for a discussion on why that might be the case.
>
> * *it remains unclear about the real usage of the regex query*
>
>     In Table 1 we have motivated regex queries based on data analysis of query logs over a publicly available KB. As KBs are highly incomplete, naive traversal cannot fully answer the queries. Therefore we need benchmarks and models to answer regex queries over incomplete KBs. Regex queries are important also because they allow a user to express many complex queries easily. For example, find the friend network of a person will require friend+, or find all ancestors of a concept in concept hierarchy can be easily expressed as hypernym+. Similarly “friend or co-worker” will require a disjunction, all employees in a person’s company will require a 2-hop path, (sibling or spouse)+ will represent all people in one’s generation related by blood or wedlock, and so on.
>
> * *does the proposed dataset contain a lot of hierarchical relations? Why would the proposed model perform better than RotatE by a lot?*
>
>     This is a good point. Indeed, FB15K contains many hierarchical relations like capitalOf → located in, headquartered in → mailing city/town, awarded an award → nominated for an award, and so on. However, it is not clear if hierarchical relations are the most prominent type of inference needed in the dataset -- probably not. We believe that a secondary reason for RotatE-Box’s better performance over RotatE is that regex queries typically have a larger answer set compared to any KBC query. And, a box makes it natural to model large sets, obtaining good performance.
>
> * *Can we get the distribution of different types of regex queries in the new two datasets?*
>
>     Kindly refer to Tables 8 and 9 (in supplementary) for this information.

---

### Decision · Program_Chairs · 2021-08-18

**Decision:**

Accept

**Comment:**

This paper present a method for performing regular expression queries (containing disjunction and Kleene plus) over KGs. A related dataset is also presented. Experimental evidence show effectiveness of the proposed method, Rotate-Box. Good contribution worthy of acceptance.